# Missing Shapiro steps in topologically trivial Josephson junction on InAs quantum well

Matthieu C. Dartiailh[1], Joseph J. Cuozzo[2], Bassel H. Elfeky[1], William Mayer[1], Joseph Yuan [1], Kaushini S. Wickramasinghe[1], Enrico Rossi [2] & Javad Shabani[1✉]

Josephson junctions hosting Majorana fermions have been predicted to exhibit a $4\pi$ periodic current phase relation. One experimental consequence of this periodicity is the disappearance of odd steps in Shapiro steps experiments. Experimentally, missing odd Shapiro steps have been observed in a number of materials systems with strong spin-orbit coupling and have been interpreted in the context of topological superconductivity. Here we report on missing odd steps in topologically trivial Josephson junctions fabricated on InAs quantum wells. We ascribe our observations to the high transparency of our junctions allowing Landau-Zener transitions. The probability of these processes is shown to be independent of the drive frequency. We analyze our results using a bi-modal transparency distribution which demonstrates that only few modes carrying $4\pi$ periodic current are sufficient to describe the disappearance of odd steps. Our findings highlight the elaborate circumstances that have to be considered in the investigation of the $4\pi$ Josephson junctions in relationship to topological superconductivity.

[1] Department of Physics, Center for Quantum Phenomena, New York University, New York, NY 10003, USA. [2] Department of Physics, William & Mary, Williamsburg, VA 23187, USA. ✉email: js10080@nyu.edu

Recently, the drive to understand and control the order parameter characterizing the collective state of electrons in quantum heterostructures has intensified. New physical behavior can emerge that is absent in the isolated constituent materials. With regards to superconductivity, this has opened a whole new area of investigation in the form of topological superconductivity[1–4]. Topological superconductors are expected to host Majorana fermions, electronic states with non-abelian statistics that can be used to realize topologically protected quantum information processing[5,6]. Topological superconductivity remains elusive in bulk materials and the majority of the research has focused on heterostructures coupling conventional superconductors to topological insulators (TIs)[7–9] or semiconducting structures with strong spin–orbit coupling in the presence of an external magnetic field[10,11].

One of the first and most important challenges to realize a topologically protected qubit is the unambiguous detection of Majorana modes. Recent experimental and theoretical works[12,13] have shown that, in general, it is very challenging to unambiguously attribute features in direct current (d.c.) transport measurements to the presence of Majorana modes. These developments show the pressing need to use alternative ways to confidently identify the presence of Majorana states and therefore of topological superconductivity.

Josephson junctions (JJs) have been proposed as a suitable platform to observe localized Majorana modes first in the context of JJs fabricated on three-dimensional (3D) TIs[2], 2D TIs[14], and, more recently, on more conventional III–V heterostructures[15,16]. JJs are attractive since the phase across the junction provides an additional knob to control the topological phase[14–16]. Among the predicted properties of JJs realized on such materials, the expected $4\pi$ periodicity of the current phase relationship (CPR) has received considerable experimental attention. The $4\pi$ periodicity of such JJs emerges from the crossing of Andreev-bound states (ABS) when the phase, $\phi$, between superconductors is equal to $\pi$. In topological JJs, formed by topological superconductors, such crossing is protected by fermion parity conservation[1,14,17]. After the crossing of the ABS's levels at $\phi = \pi$, the junction is not in the ground state anymore. As a consequence, the observation of a $4\pi$ periodic term in the Josephson current requires out-of-equilibrium alternating current (a.c.) measurements at frequencies faster than the rate of the equilibration processes[17–19]. A.c.-driven JJs can provide several features that can be associated with topological superconductivity[7–11].

A microwave drive, biasing the junction, imposes a periodic modulation of the bias current across the junction, which can lead to an advancement of the phase difference between the superconducting order parameters of the leads. When this advancement is phase locked with the dynamic of the junction, a constant voltage step, known as a Shapiro step, appears in the voltage–current (VI) characteristic of the junction[20] since the time derivative of the phase is directly proportional to the voltage. In conventional JJs, the CPR is $2\pi$ periodic[21,22]. When the leads are topological superconductors, the CPR is $4\pi$ periodic causing, in the ideal case, the absence of the odd Shapiro steps[23,24]. However, it was pointed out earlier that Landau–Zener transitions (LZTs) between ABS would lead to the same feature[25,26]. So far, no experiment has clearly and explicitly shown that this is the case.

Missing Shapiro steps have been observed in a number of topological systems such as 2D TI HgTe quantum wells (QWs)[8], 3D TI HgTe QW[7], Dirac semimetals[27], and semiconductor nanowires[10,11]. All the experiments in which missing Shapiro steps have been observed so far were designed to have the JJ in a topological state and the observation of missing Shapiro steps in those experiments is largely consistent with the presence of a topological superconducting state. Because all previous experiments were designed to observe missing Shapiro steps in a topological phase, the evidence of missing Shapiro steps, even when the system was not in a topological superconducting state, was either absent or, if present, was incidental and weak[28].

Here, we present experimental measurements that show missing Shapiro steps in high-quality JJs that are in a topologically trivial phase. The value of the results that we present is that: (i) there is no ambiguity about the fact that in our case the JJs are in a topologically trivial regime; (ii) the data presented and our analysis shows compellingly that the LZT of high-transparency modes in multi-mode junctions is a robust mechanism to obtain the fractional a.c. Josephson effect and no other mechanisms have to be present.

## Results

In this work, we present results obtained on InAs surface QWs coupled to epitaxial Al contacts. In the absence of a magnetic field, the system is topologically trivial. Still, we observe a missing first Shapiro step and a well-quantized second step similar to what has been observed in previous studies on TIs and InSb in the presence of a magnetic field[7,8,10,11]. Our results are validated by simulations including LZTs.

This study focuses on three JJs (A1, A2, and B) fabricated on two slightly different InAs surface QWs with epitaxial Al contacts. The samples are grown on semi-insulating InP (100) substrates[29–31]. The QW consists of a 4 nm layer of InAs grown on a layer of $In_{0.81}Ga_{0.19}$ as depicted in Fig. 1a. For devices A1 and A2, the capping layer is 2 nm of $In_{0.81}Al_{0.19}As$, while for device B, the capping layer is 10 nm of $In_{0.81}Ga_{0.19}As$. Figure 1b is a transmission electron microscope image of the interface between the semiconductor and the Al layer, which shows an impurity-free interface. These epitaxial interfaces are now widely used in quantum devices to study mesoscopic superconductivity[32,33], topological superconductivity[31,34], and to develop tunable qubits for quantum information technology[35]. The gap L between the superconducting contacts is 80 nm for device A1, 120 nm for A2, and 120 nm for device B, as illustrated in Fig. 1c. Details of fabrication and measurements are described in the "Methods" section.

Figure 1d presents the VI characteristic of device A1 in the absence of microwave excitation. The junction is markedly hysteretic, but as shown in a previous study and later in the text[31], this hysteresis can be ascribed to thermal effects[36] rather than capacitive effects. By fitting the linear high current part of the characteristic, we can extract the JJs normal resistance ($R_n$) and the excess current ($I_{ex}$) defined by the intersection of the fit with the x-axis. We report in Table 1 the critical current ($I_c$), along with $I_{ex}$, $R_n$, and the estimated capacitance C from a simple coplanar model[37].

From the excess current, we can estimate the average semiconductor–superconductor interface transparency of the junction estimated using the Octavio–Tinkham–Blonder–Klapwijk theory[38]. We obtain 0.85 for devices A1 and B, and 0.83 for device A2. The induced superconducting gap is taken to be one of the Al layer, which we estimate to be 220 μeV for both devices based on the critical temperature of the Al layer. We report the values of the critical current for both the cold and hot electrons branches where the cold branch goes from 0 bias to high bias and corresponds to a lower effective electronic temperature before the transition out of the superconducting state. The mean free path $l_e$ and the density have been measured on different pieces in a Van der Pauw geometry for both substrates. The density is in the range of $1 \times 10^{12}$ cm$^{-2}$ and $l_e$ is ~200 nm, meaning all presented junctions are nearly ballistic, $L < l_e$.

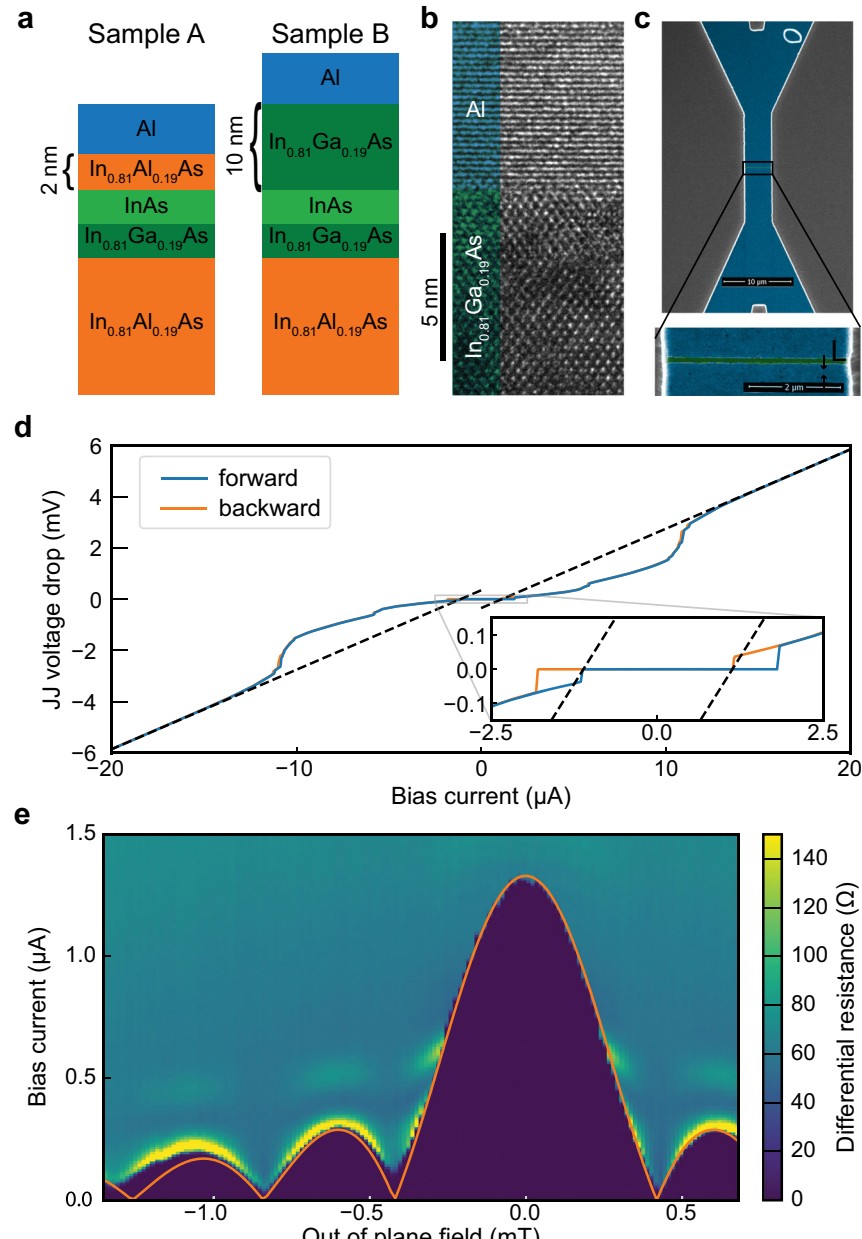

**Fig. 1 Samples structure and characterization. a** The quantum well consists of a 4 nm layer of InAs grown on top of an $In_{0.81}Ga_{0.19}As$ layer and capped with 2 nm of $In_{0.81}Al_{0.19}As$ for devices A1 and A2 and 10 nm of $In_{0.81}Ga_{0.19}As$ for device B. **b** TEM picture illustrating the expitaxial nature of the Al–InAs interface. **c** False-Color SEM image of the Josephson junction. **d** Voltage–current characteristic of device A1 in the absence of microwave irradiation. The dashed line corresponds to the linear fit at high bias used to extract the normal resistance and the excess current. **e** Differential resistance of device A1 as a function of the bias current and an out-of-plane magnetic field. The overlayed orange curve is the theoretical dependence of $I_c$ for a uniform current distribution.

**Table 1 Parameters of samples A1, A2, and B.**

|  | $I_c^{cold}$ (μA) | $I_c^{hot}$ (μA) | $I_{ex}$ (μA) | $R_n$ (Ω) | C (fF) |
|---|---|---|---|---|---|
| JJ A1 | 1.8 | 1.1 | 1.1 | 310 | ~1 |
| JJ A2 (650 mK) | 0.65 | 0.65 | 0.88 | 370 | ~1 |
| JJ B | 5.0 | 4.1 | 3.5 | 97 | ~1 |

Figure 1e is a map of the differential resistance of device A1 as a function of the bias current and an out-of-plane magnetic field. The observed Fraunhofer pattern has the expected ratio between the central lobe and the first lobe, suggesting a uniform current distribution. The period corresponds to a flux focusing yielding

an enhancement of the field by ~3, similar to values reported in ref. [39].

**Missing odd Shapiro steps.** Figure 2 presents results obtained on both devices A1 and B at two different frequencies of microwave excitation. At high frequency (namely 11 GHz for samples A1 and 12 GHz for sample B), all the expected steps are visible as can be seen from the VI characteristics presented in Fig. 2b, d. However, at lower frequencies (7 GHz for sample A1 and 6 GHz for sample B), the first Shapiro step is missing in both devices at low power. This fact is particularly clear in the VI characteristics with a microwave power of −5.7 dB (Fig. 2a) and −4 dB (Fig. 2c), respectively.

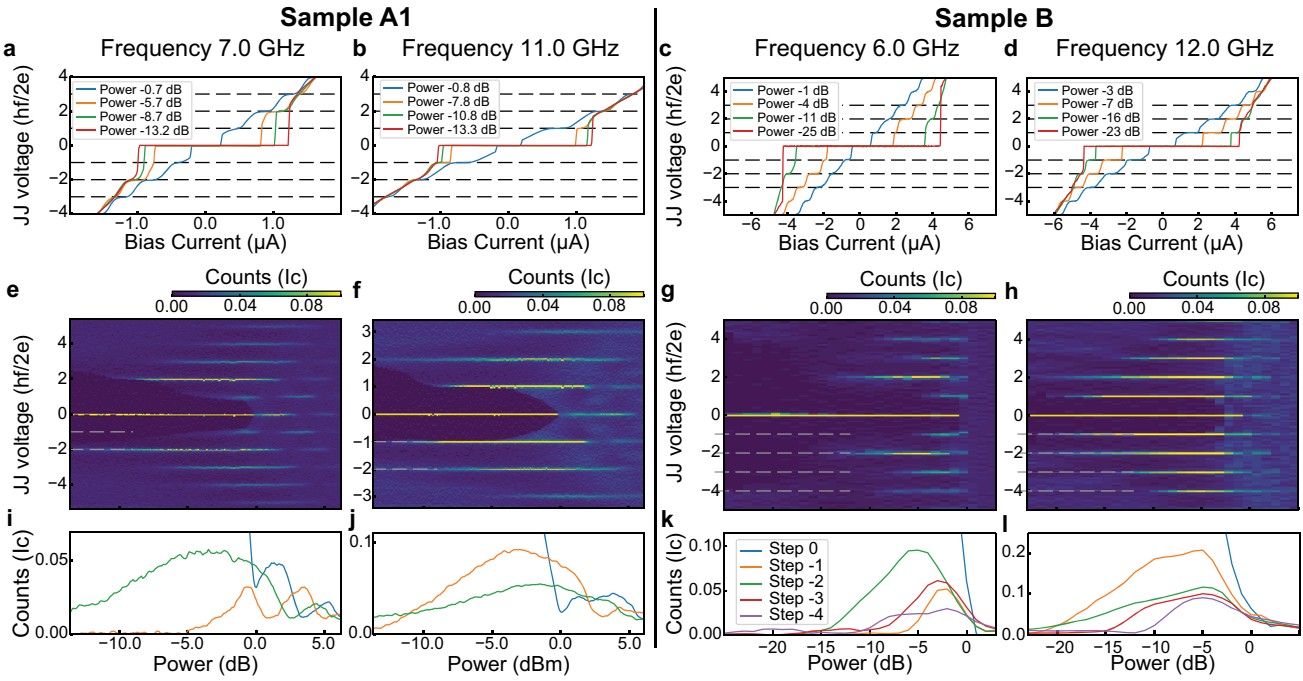

**Fig. 2 Voltage–current characteristic in the presence of microwave radiation. a–d** Histogram of the Josephson junction voltage as a function of the microwave power (**e–h**) and width of the Shapiro steps as a function of the microwave power (**i–l**) for devices A1 and B at two different frequencies for each device. In both the histogram and the plot of the width of each step, counts are expressed in unit of the critical current of the junction in the absence of the microwave drive. In both devices, while at high frequencies all Shapiro steps are visible, at low frequency and low power, the first step appears to be strongly suppressed. The microwave power is normalized to the power required to observe the vanishing of the critical current.

In order to present the power dependence of the Shapiro steps, we plot, for each device and frequency, the histogram of the voltage distribution as a function of the microwave power in Fig. 2e–h. Finally, in Fig. 2i–l, we plot the weights of the steps as a function of power. As seen already in the VI characteristics, for each sample the first step is strongly suppressed at low frequency and at low power. In addition, in the case of sample B, a weak suppression of the third Shapiro step is visible in Fig. 2g, k. This signature is also present in the data presented in Supplementary Fig. 1 at 4 and 5 GHz on the same sample, allowing us to rule out spurious experimental setup effects.

**Experiments at a higher temperature**. As already mentioned, our samples present a marked hysteresis that can be ascribed to local heating effects. Such a hysteretic behavior could lead to the disappearance of the first Shapiro step for trivial reasons. On the cold branch, the system can stay trapped in the superconducting state up to current values far exceeding the value of $I_c^{hot}$ governing the Shapiro steps physics. As a consequence, when the system makes a transition out of the superconducting state, a large voltage step occurs. On the hot branch, the system can abruptly jump from a finite voltage to the superconducting state. In both cases, the sudden voltage jump could lead to the observation of one or more missing Shapiro steps as strikingly observed recently in ref. [40]. Such a mechanism cannot suppress the third step without affecting the second one, as a consequence, we focus the following discussion on device A1.

In device A1, the re-trapping on the hot branch occurs at a voltage of 36 μV. We observe a single missing Shapiro step down to 4 GHz as shown in Supplementary Fig. 1a, which corresponds to a voltage of ~8.3 μV. If the hysteresis effects constituted the main suppression mechanism of the Shapiro steps, we could hence expect to also see other steps missing.

To confirm that our observations cannot be simply attributed to hysteresis in the JJ, we have performed additional measurements at a higher temperature for which the critical current is lower and the junction does not exhibit hysteresis. These measurements have been performed on device A2, which was fabricated on the same piece of a wafer as device A1, but with a gap L of 120 nm. We first present, in Fig. 3a, the temperature dependence of the $I_c R_n$ product of device A2 on the cold and the hot branches. We observe that the value of the product on the hot branch is globally constant up to 600 mK, while the cold branch value decreases, which is expected when the hysteresis is of thermal origin[36]. At 650 mK, both curves overlap and the system is hysteresis free as illustrated in Fig. 3b. We will hence focus on this temperature for the rest of the discussion on device A2.

Figure 3c, d present the VI characteristic of device A2 in the presence of microwave radiation at 4.2 and 11.5 GHz, respectively. The data now present a quasi-continuous variation of the voltage, and as seen in device A1 and B, the first Shapiro step appears to be missing at the lowest frequency. This observation is confirmed when looking at the histograms of the VI characteristic and the dependence of the Shapiro steps width on power presented in Fig. 3e–h The results of Fig. 3 show that the lowest odd Shapiro step is missing even when no hysteresis is present.

Overall, our experimental observations of missing Shapiro steps are quite similar to the ones obtained on platforms expected to host Majorana modes, but in our case, the system is trivial in absence of a magnetic field. We note that similar InAs QW may host a topological phase[31,34], in the presence of a sizable in-plane magnetic field. Data presented here are all taken at zero magnetic field and hence our JJs are topologically trivial.

**Theoretical analysis**. A microscopic analysis of the dynamics of the JJ taking into account the presence of all the transverse modes, and a biasing current with both a d.c. and a.c. terms is

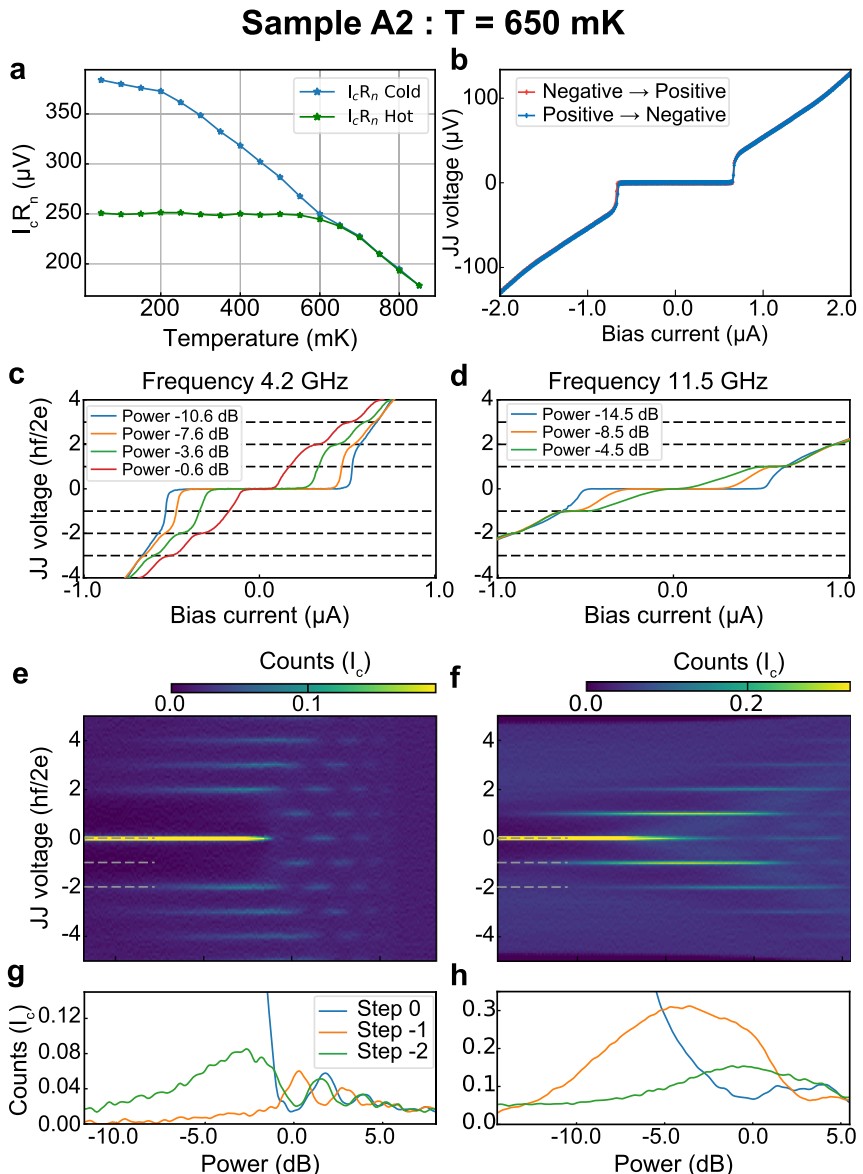

**Fig. 3 Control experiment at elevated temperature. a** Dependence of the $I_cR_n$ product of device A2 on both the cold and hot branches as a function of temperature. **b** VI characteristic of device A2 at 650 mK. Voltage–current characteristic in the presence of a microwave radiation (**c, d**), histogram of the Josephson junction voltage as a function of the microwave power (**e, f**), and width of the Shapiro steps as a function of the microwave power (**g, h**) for samples A2 at 650 mK and a microwave drive of 4.2 and 11.5 GHz. The normalization follows the same conventions used in Fig. 2.

computationally prohibitive[41]. For this reason, we describe the JJ via an effective shunted junction model. In general, the model has both a resistive and a capacitive channel. The junctions considered in this work, given their geometry, have a very small capacitance $C \sim 1$ fF. We, therefore, can neglect the capacitive channel and model the JJ's dynamics using a resistively shunted junction model:

$$\frac{\hbar}{2eR_n}\frac{d\phi}{dt} = I_{d.c.} + I_{ac}\sin(2\pi f_{a.c.}t) - I_s(\phi). \quad (1)$$

where $I_{d.c.}$ is the d.c. bias current, $I_{a.c.}$ the amplitude of the a.c. current due to the microwave radiation with frequency $f_{a.c.}$, and $I_s(\phi)$ is the supercurrent. For modes with transparency $\tau$, the probability of an LZT at $\phi = \pi$ is given by[42],

$$P_{LZT}(t) = \exp\left(-\pi\frac{\Delta(1-\tau)}{e\,|V(t_{LZT})|}\right), \quad (2)$$

with $V(t_{LZT}) = (\hbar/2e)d\phi/dt|_{\phi=\pi}$ the instantaneous voltage across

the junction when the system goes across the anti-crossing at $\phi = \pi$ in the spectrum.

To model the supercurrent flowing across the JJ, we use two effective modes: a very low transparency mode with a purely sinusoidal CPR and negligible probability to undergo an LZT, and an effective mode with very high transparency $\tau$, which can undergo an LZT at $\phi = \pi$. This is justified considering the spectrum of the ABS for the type of JJs that we study.

Let $n_y$ be the integer labeling the transverse modes. For each transverse mode, $n_y$, the longitudinal Fermi velocity $v_{F,x}^{(n_y)}$ is given by:

$$v_{F,x}^{(n_y)} = \frac{2at}{\hbar}\left(\frac{\mu}{t} - \pi^2\frac{a^2}{W^2}n_y^2\right)^{1/2}, \quad (3)$$

where $t = \hbar^2/(2m^*a^2)$ is the nearest-neighbor hopping amplitude, with $m^*$ the effective mass, $a$ the lattice constant, and $\mu$ the chemical potential. We have $m = 0.03\,m_e$[29], $a = 0.6$ nm, and

$\mu \approx 0.1$ eV. From $v_{F,x}^{(n_y)}$ we can then extract the effective "longitudinal" coherence length of each transverse mode, $\xi_{n_y} = \hbar v_{F,x}^{(n_y)}/(\pi\Delta)$. For $W = 4\,\mu$m and $L = 100$ nm, the typical length and width of the JJs studied here, we obtain that there are few modes for which $L > \xi_{n_y}$, that is, modes for which the junction appears to be long. As shown in Supplementary Fig. S4, the ABS corresponding to these modes will thus have energy well below $\Delta$ for any value of $\phi$. In our devices, these modes are also highly transparent and therefore have a very high probability to undergo an LZT at $\phi = \pi$. In addition, given their large separation from the continuum, they have negligible probability to undergo LZTs to the continuum. These modes are therefore responsible for the $4\pi$ component of the CPR. This situation is analogous to the one discussed in ref. [43], in which a resonant impurity-bound state, by weakly coupling to a high transparent mode, effectively turned it into a "long-junction" mode well separated from the continuum at $\Delta$. The modes for $L < \xi_{n_y}$ have lower transparency and so lower probability to undergo an LZT for $\phi = \pi$. In the presence of disorder, the "long-junction" modes become diffusive. In this case, given the bimodal character of distribution of transparencies in diffusive conductors[44–46], we expect that some of the "long-junction" modes will still have transparency close to 1.

One might expect that for some modes, the $P_{LZT}$ when $\phi = \pi$ might be small but not negligible so that they could affect qualitatively the Shapiro steps. To check for this, we studied the effect of different values of the transparency for the two channels model that we present. For intermediate transparencies, we obtained a chaotic dynamic for the phase that resulted in no discernible Shapiro steps. We can attribute the inability of modes with intermediate transparency to contribute terms to the CPR with periodicity different from $2\pi$ to the fact that these modes have a very large probability to merge into the continuum for $\phi = 2n\pi$ ($n \in \mathbb{N}$) and therefore are very unlikely to contribute a $4\pi$ term to the CPR even if they undergo an LZT at $\phi = \pi$. In addition, the combined effect of disorder and possible phase fluctuations of the order parameter along the transverse direction —that cannot be excluded considering that the width of the junctions (in our samples $4\,\mu$m) is much larger than the superconducting coherence length—will tend to suppress the probability of LZTs at $\phi = \pi$ for modes with intermediate-low transparency. The presence of highly transparent modes "long-junction" seems to be the key for the observation of missing Shapiro steps in our devices.

Considering the two effective modes, the total supercurrent $I_s$ across the junction is given by:

$$I_s(\phi) = I_c\left( s\frac{1-n}{\alpha_\tau}\frac{\sin(\phi)}{\sqrt{1-\tau\sin^2(\phi/2)}} + \frac{n}{\alpha_0}\sin(\phi)\right), \quad (4)$$

where $I_c$ is the experimentally measured critical current, $n$ is the fraction of the current from the low transparency mode, and $\alpha_\tau$ and $\alpha_0$ are the values of $\sin(\phi)/\sqrt{1-T\sin^2(\phi/2)}$ and $\sin(\phi)$, respectively, for the angle $\phi$ for which the current is maximum. We neglect interference effects due to phase fluctuations and coherence between LZTs[26]. We do not consider relaxation from higher to lower energy states, which we expect to be suppressed. In particular, relaxation between those states through the continuum[47] should be absent since the modes carrying a $4\pi$ supercurrent are well separated from the continuum as discussed above and shown in Supplementary Fig. 4. By solving Eq. (1) accounting for the dynamics due to the LZTs of the effective high-transparency mode, we obtain the time evolution of $\phi$, and then the d.c. voltage by calculating the time average, $\overline{V}$, of $V(t)$.

Figure 4a, b shows the dynamics of the phase and the instantaneous voltage in the second Shapiro step using the parameters of device A1, which means that $hf_{ac} \ll hf_J = 2eR_nI_c$. Even though there is a factor of 2 between the drive frequency employed in each case, the value of the instantaneous voltage $V(t)$ when $\phi = \pi$, that is, $V(t_{LZT})$, is the same in both cases as indicated by the black stars in Fig. 4a, b. This conclusion can be recovered by considering the analogy between the dynamics of the phase, described by Eq. (1), and the damped dynamics of a massless (given that the capacitance is negligible in our devices) particle in a washboard potential. The instantaneous voltage across the junction spikes when the particle falls from one minimum of the washboard to another. This happens when the sum of $I_{d.c.}$ and $I_{a.c.}$ overcome the modulation of the washboard potential ($I_c$). The speed the particle achieves during this transition is directly related to the friction ($R_n$), and amplitude of the d.c. and a.c. component of the current. In the first steps, the sum of both those amplitude is typical of the order of the modulation of the washboard potential ($I_c$), which they need to overcome to induce a movement of the particle. It is, however, independent of the frequency since the particle is massless. We provide additional numerical results supporting this conclusion in Supplementary Figs. 6 and 7.

This result is important since the lower frequency at which missing steps are observed has been used to estimate the required transparency for LZTs to explain missing steps[7]. In the presence of LZT, Shapiro steps are well quantized only if the probability of the transition is very close to 0 or 1 as shown in ref. [7] and Supplementary Fig. 9. In all three devices, the induced superconducting gap is close to the bulk gap of the Al layer $\Delta = 220$ μeV = 53 GHz, given the high interface transparency, and the lowest frequency at which we observe Shapiro steps is 4 GHz. Using the value of the voltage corresponding to the lowest frequency for which the first odd Shapiro step is missing and LZT probability of 0.97, we would get that the transparency of the mode undergoing LZTs should be >0.9985. However, using the value $V(t_{LZT})$ shown in Fig. 4a, b using black stars, that is, $V(t_{LZT}) = 1.8\Delta/e$, we obtain that a transparency of 0.982 is sufficient.

In the presence of both $2\pi$ and $4\pi$ periodic components in the CPR, odd steps are expected to be suppressed when the a.c. drive frequency is lower than $f_{4\pi} = 2eR_nI_{4\pi}/h$[23,24]. For devices A1 and B, $f_{4\pi}$ can be estimated to be ~10 GHz, yielding $I_{4\pi} \sim$ 7 nA for device A1 and 260 nA for device B, which in both devices corresponds to ~6% of the critical current on the hot branch. For device A2, the limit frequency is ~6.5 GHz, as shown in Supplementary Fig. 2, yielding an $I_{4\pi} \sim$ 35 nA, which corresponds to 5% of $I_c$ ($T = 650$ mK). Using the Al gap value for the induced gap, we can estimate the amount of current carried by a single mode $I_{mode} = \frac{e\Delta}{2\hbar} \sim 25$ nA. This means that the total $4\pi$ periodic contribution to the CPR can be assigned to ~3 (1.5, 10) modes for the device A1 (A2, B). The typical densities reported for each sample yield between 320 and 550 transverse modes in each JJ. This means that only a small minority of the modes (0.5–1% in device A1, 0.25–0.5% in device A2, and 2–3% in device B) need to have near unity transparency and participate in LZT processes to explain our observations.

Figure 4c–f presents the simulation results for both device A1 and B for the same frequencies presented in Fig. 2. For device A1, panels c and e, we assumed that 5% of the current is carried by modes that have a transparency $\tau = 0.98$. For device B, panels d and f, we assumed that just 3% of the current is carried by modes that have such high transparency. For both devices, the fraction of the current associated with high-transparency modes is very close to the values extracted from the value of the threshold frequency

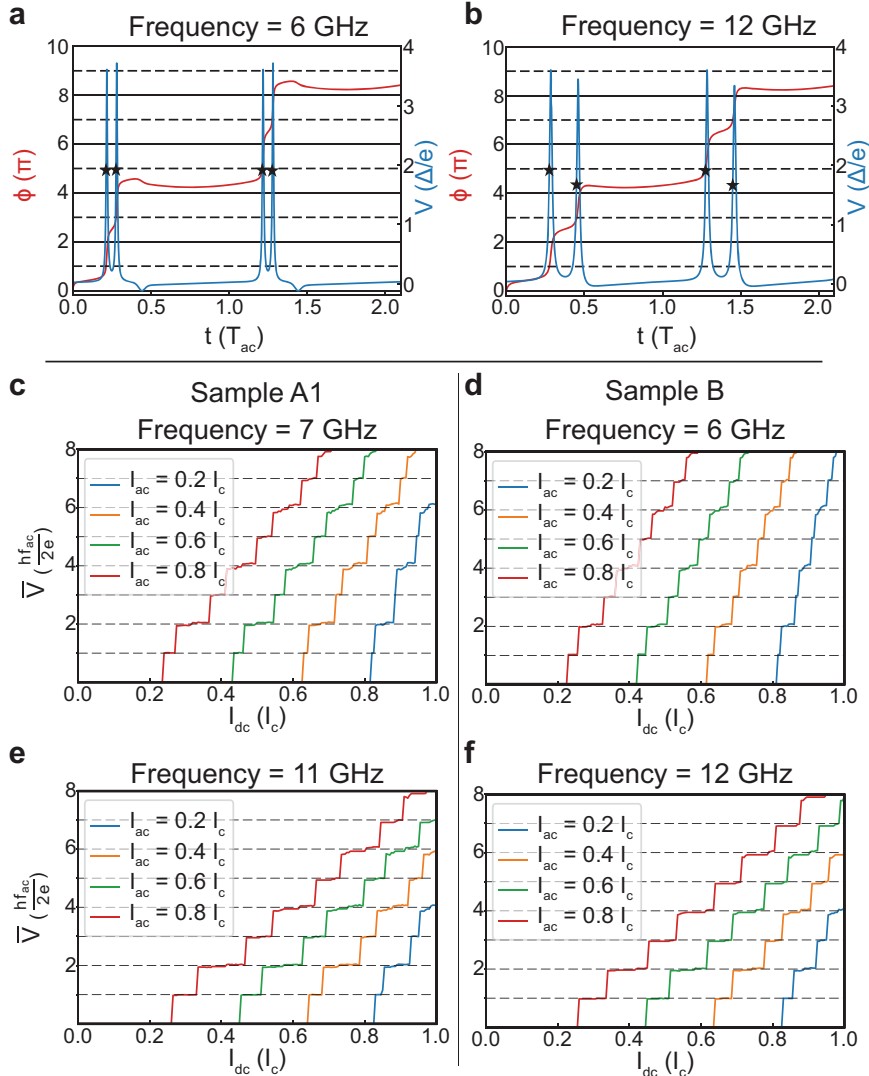

**Fig. 4 Resistively shunted junction (RSJ) simulation accounting for LZTs. a, b** Phase and the instantaneous voltage across the JJ. The parameters of sample B are used with $\tau = 0.98$ and $n = 0.97$. The driving currents are $I_{d.c.} = 0.7I_c$ at 6 GHz and $0.8I_c$ at 12 GHz, and $I_{a.c.} = 0.4I_c$ at both frequencies, which corresponds to the second Shapiro step. The black stars indicate the instantaneous voltage at $\phi = \pi$ used in the calculation of the Landau–Zener transition. **c, e** $\overline{V}$–$I_{d.c.}$ curves showing Shapiro steps at 7 and 11 GHz for the parameters of device A1, with $\tau = 0.98$ and $n = 0.95$. **d, f** $\overline{V}$–$I_{d.c.}$ curves showing Shapiro steps at 6 and 12 GHz for the parameters of device B, with $\tau = 0.98$ and $n = 0.97$.

above which the first Shapiro steps reappears. The results of Figs. Fig. 4c–f show that the two-channel model, in which just a small fraction of the current is carried by high-transparency modes with effective coherence length smaller than the junction length, is able to reproduce well the experimental results, and, in particular, the missing first Shapiro steps in topologically trivial Josephson junctions.

## Discussion

In this work, we show experimentally that in JJs that are undoubtedly in a topologically trivial phase, for the microwave powers and frequencies reported, there are missing odd Shapiro steps consistent with the $4\pi$ periodic current phase relation of a topological JJ. We attribute our measurement to the very high transparency of a fraction of the modes in our JJ combined with large value of $I_cR_n$. Our results clearly show that caution should be used to attribute missing Shapiro steps to the presence of Majorana modes. They provide essential guidance to future experiments to use JJs to unambiguously establish the presence of

topological superconductivity, and, in addition, significantly enhance our understanding of high-quality JJs.

## Methods

**Fabrication.** The samples were grown on semi-insulating InP (100) substrates in a modified Gen II MBE system. The step graded buffer layer, $In_xAl_{1-x}As$, is grown at low temperature to minimize dislocations forming due to the lattice mismatch between the active region and the InP substrate. After the QW is grown, the substrate is cooled to promote the growth of epitaxial Al (111)[29–31]. The typical thickness of the grown aluminum layer is 20 nm.

The fabrication process consists of two steps of electron beam lithography using poly(methyl methacrylate) resist. After the first lithography, the deep semiconductor mesas are etched using first Transene type D to etch the Al and then a III–V wet etch ($C_6H_8O_7$ (1M) 18.3:$H_3PO_4$ (85% in mass) 0.43:$H_2O_2$ (30% in mass) 1:$H_2O$ 73.3, the volumes are normalized to the volume of $H_2O_2$ used) to define deep semiconductor mesas. The second lithography is used to define the JJ gap that is etched using Transene type D.

**Measurements.** Devices are measured in a four-point geometry using a current bias configuration. Differential resistance is measured using a lock-in amplifier SRS860. The a.c. signal used has an amplitude of 10 nA and a typical frequency of 17 Hz. For measurements carried out on devices A1 and A2, a differential amplifier

NF-SA440F5 is used to measure the voltage drop across the JJ. Microwave excitation is provided through a nearby antenna. All measurements are performed in a dilution fridge with a mixing chamber temperature of 30 mK.

## Data availability

The data that support the findings of this study are available within the paper and its Supplementary information. Additional data are available from the corresponding author upon request.

## Code availability

The numerical simulation code is available from the corresponding author upon request.

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

## Acknowledgements

NYU team is supported by NSF DMR Grant No. 1836687 and DARPA Grant No. DP18AP900007. J.Y. acknowledges funding from the ARO/LPS QuaCGR fellowship. J.J.C. and E.R. acknowledge support from ARO Grant No. W911NF-18-1-0290 and NSF CAREER Grant No. DMR-1455233. J.J.C. acknowledges research supported from the Graduate Research Fellowship awarded by the Virginia Space Grant Consortium (VSGC). W.M. team acknowledges William & Mary Research Computing for providing computational resources and/or technical support that have contributed to the results reported within this paper. URL: https://www.wm.edu/it/rc.

## Author contributions

W.M., M.C.D., and B.H.E. fabricated the devices and performed the measurements with J.S. providing input. J.Y., K.S.W., and J.S. designed and grown the epitaxial Al/InAs heterostructures. M.C.D. and B.H.E. performed data analysis and J.J.C. and E.R. developed the theoretical model and carried out the simulations. J.S. conceived the experiment. All authors contributed to interpreting the data. The manuscript was written by M.C.D., J.J.C., E.R., and J.S. with suggestions from all other authors.

## Competing interests

The authors declare no competing interests.
