## [Peer Review File · Nature Communications]

Reviewer #1 (Remarks to the Author):

Dear Editor,

The manuscript titled "Missing Shapiro steps in topologically trivial Josephson Junction on InAs quantum well" provides detailed measurements of missing Shapiro steps in InAs quantum wells with no magnetic fields.

These results provide evidence that supports theoretical works that suggested that LZT between ABSs could lead

to the such missing steps. The crux of the originality of this work relies on the claim

"So far, no experiment has shown that in realistic setups this is the case and all the experiments in which miss

ing Shapiro steps have been observed are consistent with the presence of a topological superconducting state".

While I agree with the statement in the formal sense that all papers with missing Shapiro steps have claimed

to be in the topological superconducting state, below I argue that they also contained strong evidence of not

being in the topological state.

As an example of missing Shapiro steps in a non-topological state, consider Fig 4c of Ref. 13 in the manuscript. While the data presented is on HgTe, which can be in the topological regime in a way similar to

the material in the manuscript in a magnetic field, the data in Ref 13 is in the so-called Fraunhofer gate voltage regime. This regime, in other words is also called the n-doped regime where the HgTe is no longer a topological insulator with edge states but rather an n-doped semiconductor. This regime is not topological in any conventional sense, though Ref 13 certainly seems to ignore this point.

Even closer to the experimental system in the manuscript is Pribyl et al (Nat. Nano. 10, pages 593–597(2015)),

which studies JJ of InAs-InSb 2DEGs. If we look at Fig 3c, the caption states that in the n-region all Shapiro steps are present. However, the statement is only on a particular high value of power.

Looking

at the figure (as was pointed to me by authors of the papers in private communications), at low powers,

the first step would indeed be missing for positive bias currents. Both these examples and two other examples

were pointed out in the introduction of a theory paper PRB 95, 060501(R) (2017) as examples of cases where

LZTs were likely playing a role. Therefore, I think a more nuanced motivation for this work is needed highlighting the connection to LZT transitions strengths etc.

In addition, I think there are a few citation issues in the manuscript. The 4π periodic Josephson current

is alluded to at the end of Kitaev's pioneering 2001 Majorana wire paper. The manuscript implicitly credits this

to Ref 1 in the beginning of the paper. In fact Ref 20 discussed this in a more realistic context well before

Ref 1. Additionally the authors state "However, it was pointed

out earlier that Landau-Zener transitions (LZT) between Andreev bound states would lead to the same feature"

without any reference to what "earlier" means.

In terms of presentation an important point that requires further clarification is the statement on page

4

"the value of the instantaneous voltage $V(t)$ at π is the same...phase locked to the a.c drive." This is because naively phase locked to the ac drive implies that the sweep rate would be related to frequency.

It would be helpful if the authors could expand on what they precisely mean on page 5 by "intermediate transparencies...contributions to LZT experiments." Did the authors do a multi-channel modeling?

Finally, it would be nice if the authors could clarify the role of LZT. This is because highly transparent junctions can produce LZT at both zero energy as well as energy Δ (see e.g. PRB 95, 060501(R) (2017)), which then destroys the 4π periodic nature of the current.

In summary, I think the present manuscript can be interesting as a systematic study of non-topological origin of missing Shapiro steps. At the same time, because of the muddled nature of the literature I cannot accept the claim that this is the first time that this has been seen, though it hasn't been stated before. In my view, whether this manuscript should be published in Nature Comm or a more specialized journal depends on the extent to which it can clarify this issue and characteristics of Shapiro from non-topological sources.

Reviewer #2 (Remarks to the Author):

The authors report the realization of Al-InAs-Al heterostructure using MBE epitaxial deposition. They measured the d.c. and a.c. the response of the samples. A surprising suppression of the odd-number Shapiro steps was observed, which the authors attributed to the Landau-Zener transition in few Andreev bound states with high transparency. I believe that such good samples and data quality worth being published. But before I fully recommend the publication, there are few points that need to be addressed in the revised version.

1. The Landau-Zener transition probability in a highly transparent ABS is simply given by $P_{LZT} = \exp(-\pi\Delta(1-\tau)/eV)$ (Equ. (3) in the manuscript) which suggests a higher transition probability at higher bias voltage, similarly at a higher frequency. This should suggest a stronger 4π signal (suppression of the odd steps), which is contradictory to the observed results (at higher bias voltage, the odd missing steps reappear). This was also argued in the previous works by showing a frequency dependence and the recovered odd steps at higher bias voltage. Could the authors provide an explanation within the same model?

2. One reason that the 4π -periodic current was not observed in many highly transparent trivial systems is that the quasiparticle poisoning rate (states drop from the upper ABS branch to the lower one) is much higher in these systems. It is essential to understand why it is significantly lower in the reported system. A major cause of such relaxation is the inelastic scattering, which can be strongly suppressed in a system with strong spin-orbit coupling. In ref.[27], authors point out that two zero-energy modes can be formed due to two degenerate Dirac cones with opposite chirality, and the orthogonality between the two cones prevents the two modes from annihilating. Therefore a $4n$ -periodic current can be observed. A similar picture can be found in the current InAs junction, without the magnetic gap, two Fermi surfaces at different momenta should have a low inter-cone scattering rate, and give two (very near) zero-energy modes. Can the authors elaborate more on this in the discussion part?

3. Please provide details of the fabrication process. E.g. the thickness of the Al layer, how many lithography steps were involved, etc.. The authors refer to the supplementary information as "Details of fabrication and measurements are described in the SI.", but there is no description in the SI.

Reviewer #3 (Remarks to the Author):

In the manuscript by Dartiailh, et al., the authors study Al/InAs/Al Josephson junctions under microwave irradiation. They map out the patterns of Shapiro steps in the current-voltage characteristics of such devices as a function of microwave power and find that the first voltage step is suppressed at low RF drive amplitude. While a similar suppression has been observed in Josephson devices with weak links made from topologically nontrivial materials, the devices under test are in the trivial regime. The authors attribute the effect to Landau-Zener transitions between Andreev bound states of channels with high transparency and go on developing a RSJ model description of the junction dynamics. They add the contribution of a single highly transparent mode to an otherwise sinusoidal current-phase relation and allow for Landau-Zener transitions between the two states of the transparent mode at odd multiples of phase π with a phase velocity dependent LZ probability. (I infer this from the reference by Dominguez, et al., since it is not clear in the text.) It is then found from the numerical solutions of the RSJ model that Shapiro steps at voltages corresponding to odd multiples of the drive frequency are suppressed at low drive amplitude (most clearly seen in Fig. S2). Based on these data the authors argue the case for 4π -periodic dynamics due to Landau-Zener transitions in their junctions. The appearance of weak half-integer Shapiro steps at higher drive frequencies is presented as additional evidence for the presence of high transmission modes in these samples.

Shapiro step patterns with missing steps have been put forward as key evidence for (topological) midgap Andreev bound states in Josephson devices with topological insulator weak links. Critics have argued that it should be easy to replicate this effect in topologically trivial Josephson devices with (few) highly transparent modes if Landau-Zener transitions occur between the two states. I believe it to be of great value to verify such a scenario experimentally, but I am not aware of any experimental results. Dartiailh, et al., report that the first Shapiro step in the current-voltage characteristics of Al/InAs/Al Josephson junctions is missing at low microwave drive amplitude. They attribute this to a 4π -periodic Josephson effect enabled by Landau-Zener transitions. However, I find the data unconvincing. Hysteresis effects can easily be mistaken for a missing first Shapiro step. Although the authors state that this is not the case, I must disagree.

Firstly, from the tiny insert in Fig 1d, the voltage at which retrapping occurs in sample A is ca. 25 μV . This corresponds to a frequency of ~ 12 GHz. Below this frequency the hysteresis interferes with the visibility of the first Shapiro step. Secondly, despite the unfortunate choice of a color scheme, we can see this directly in the outlines of the dark blue zero-voltage region at low microwave power in the colorplot, Fig. 2e/f (sample A). From the parameters in Table 1 and colorplot Fig 2g/h, I presume that the situation is similar in sample B. Additionally, the authors point out a weak suppression of the third step in Fig 2g/k. Here I urge caution because resonances in the device or measuring circuit wiring may wash out one or more of the steps. The Shapiro step patterns should be studied carefully for several (nearby) frequencies and in a larger range of microwave powers to rule out such spurious effects.

In my opinion, the experimental data in the manuscript do not support the conclusion. To make a stronger case, additional measurements are needed. The authors could, e.g., shunt the device with a parallel resistor to remove the hysteresis or repeat the measurements at higher temperatures when the critical current is smaller. I cannot recommend publication of the manuscript in its present form.

Nature Communication: Answer to the referees

Below we show in blue the referees' comments and in black our reply. We also include a summary of the changes made to the main text and the Supplementary Information at the end of our answer.

I. REFEREE 1: REMARKS TO THE AUTHORS

We thank the referee for the thoughtful comments and for pointing out some of the shortcomings of the analysis presented in the original version of the manuscript. Below we address in detail the issues raised by the referees and briefly describe the changes made in the revised version of the manuscript to address these issues.

The manuscript titled "Missing Shapiro steps in topologically trivial Josephson Junction on InAs quantum well" provides detailed measurements of missing Shapiro steps in InAs quantum wells with no magnetic fields. These results provide evidence that supports theoretical works that suggested that LZT between ABSs could lead to the such missing steps. The crux of the originality of this work relies on the claim "So far, no experiment has shown that in realistic setups this is the case and all the experiments in which missing Shapiro steps have been observed are consistent with the presence of a topological superconducting state". While I agree with the statement in the formal sense that all papers with missing Shapiro steps have claimed to be in the topological superconducting state, below I argue that they also contained strong evidence of not being in the topological state.

As an example of missing Shapiro steps in a non-topological state, consider Fig 4c of Ref. 13 in the manuscript. While the data presented is on HgTe, which can be in the topological regime in a way similar to the material in the manuscript in a magnetic field, the data in Ref 13 is in the so-called Fraunhofer gate voltage regime. This regime, in other words is also called the n-doped regime where the HgTe is no longer a topological insulator with edge states but rather an n-doped semiconductor. This regime is not topological in any conventional sense, though Ref 13 certainly seems to ignore this point. Even closer to the experimental system in the manuscript is Pribyl et al (Nat. Nano. 10, pages593–597(2015)), which studies JJ of InAs-InSb 2DEGs. If we look at Fig 3c, the caption states that in the n-region all Shapiro steps are present. However, the statement is only on a particular high value of power. Looking at the figure (as was pointed to me by authors of the papers in private communications), at low powers, the first step would indeed be missing for positive bias currents. Both these examples and two other examples were pointed out in the introduction of a theory paper PRB 95, 060501(R) (2017) as examples of cases where LZTs were likely playing a role. Therefore, I think a more nuanced motivation for this work is needed highlighting the connection to LZT transitions strengths etc.

We agree with the referee's general observation that in some previous experiments in which missing Shapiro steps were attributed exclusively to the presence of a topological superconducting state, some evidence could be found that in some regime the missing Shapiro steps could be attributed to other mechanisms. However, because all previous experiments were designed to observe missing Shapiro steps in a topological phase, we would argue that the "side" evidence of missing Shapiro steps even when the system was not in a topological superconducting state was not very strong,

and not as strong as the results we present, particularly after taking into account the additional data that we have included in the revised manuscript motivated by the referees' comments.

In general, to infer from the references mentioned by the referee that the missing Shapiro steps are not due to the presence of a topological superconducting state requires a careful analysis of the data presented in those references and possibly private communications with the authors of such works, as the referee mentions in his/her report. And even then the situation can be unclear. In particular, in the case of HgTe, there are both theoretical predictions (*X. Dai, T. L. Hughes, X.-L. Qi, Z. Fang, and S.-C. Zhang, Helical Edge and Surface States in HgTe Quantum Wells and Bulk Insulators, Phys. Rev. B 77, 125319 (2008)*) and experimental works (*K. C. Nowack, et al., Imaging Currents in HgTe Quantum Wells in the Quantum Spin Hall Regime, Nature Materials 12, 787 (2013)* and *M. C. Dartiailh, et al., Dynamical Separation of Bulk and Edge Transport in HgTe-Based 2D Topological Insulators, Phys. Rev. Lett. 124, 076802 (2020)*) suggesting that the topological edge states can survive with the bottom of the conduction band making it hard to assign missing Shapiro's step to topological states or highly transparent channels. The value of the work that we present is that:

- There is no ambiguity about the fact that in our case the Josephson junctions are in a topologically trivial regime;
- The data presented show compellingly that Landau-Zener transition (LZT) in high-transparency, multi-mode, junctions is indeed a robust mechanism to obtain the fractional ac Josephson effect and no other mechanisms have to also be present.

We agree that the theory paper, PRB 95, 060501(R) (2017), (Ref. 43 in the revised manuscript) mentioned by the referee provides a useful framework to understand how it is possible to have missing Shapiro steps even when the Josephson junction is not topological. The Josephson junctions that we study are very wide and therefore the spectrum of the Andreev bound states (ABSs) has several modes due to the large number of transverse modes in the junction. As we explain more in detail when addressing another one of the reviewer's comments below, there is an analogy between the ABS spectrum considered in Ref. 43, and the ABS's spectrum of the devices that we study that is key for the observation of missing Shapiro steps.

In the revised manuscript to address this important issue we explained in more accurate terms the importance of our work in the context of the existing literature on missing Shapiro steps in topological Josephson junctions. Motivated also by the comments of referee #3 in the revised manuscript we also present new results, see Fig. 3, in which no hysteresis effects are present and so we are able to show compellingly that the LZTs of the high transparency modes are responsible for the fractional ac Josephson effect. In the revised manuscript we have also expanded the theoretical analysis and pointed out the connection between the theoretical treatment presented in Ref. 43 and the mechanism that we conclude is responsible for the fractional ac Josephson effect in our devices.

In addition, I think there are a few citation issues in the manuscript. The 4pi periodic Josephson current is alluded to at the end of Kitaev's pioneering 2001 Majorana wire paper. The manuscript implicitly credits this to Ref 1 in the beginning of the paper. In fact Ref. 20 discussed this in a more realistic context well before Ref 1. Additionally the authors state "However, it was pointed out earlier that Landau-Zener transitions (LZT) between Andreev bound states would lead to the same feature" without any reference to what "earlier" means.

The referee is correct. In the revised manuscript when we first discuss the 4π periodic Josephson current we refer to the paper Kitaev and the paper by Kwon et al. (Refs. 1 and 2 in the revised manuscript), and then to the paper by Fu and Kane. When discussing Landau-Zener transitions as a mechanism that can give rise to a 4π periodic Josephson current, in the revised manuscript we have added references 25 and 26: *Billangeon, P.-M., Pierre, F., Bouchiat, H. & Deblock, R. ac Josephson Effect and Resonant Cooper Pair Tunneling Emission of a Single Cooper Pair Transistor. Phys. Rev. Lett. 98, 216802 (2007).* and *Domínguez, F., Hassler, F. & Platero, G. Dynamical detection of Majorana fermions in current-biased nanowires. Phys. Rev. B 86, 140503 (2012).*

In terms of presentation an important point that requires further clarification is the statement on page 4 "the value of the instantaneous voltage $V(t)$ at π is the same...phase locked to the a.c drive." This is because naively phase locked to the ac drive implies that the sweep rate would be related to frequency.

The fact that $V(t)$ at $\phi = \pi$ (ie $V(t_{LZT})$) is fairly insensitive to the value of the ac frequency, in the limit $hf_{ac} \ll eR_n I_c$, can be recovered by considering the analogy between the dynamics of the phase and the damped dynamics of a massless (given that the capacitance is negligible in our devices) particle in a washboard potential. The instantaneous voltage across the junction spikes when the particle falls from one minimum of the washboard to another. This happens when the sum of I_{dc} and I_{ac} overcome the modulation of the washboard potential (I_c). The speed the particle achieves during this transition is directly related to the friction (R_n), and amplitude of the dc and ac component of the current. In the first steps the sum of both those amplitude is typically of the order of the modulation of the washboard potential (I_c), which they need to overcome to induce a movement of the particle. It is, however, independent on the frequency since the particle is massless.

In the revised manuscript we have added a paragraph similar to the one above to explain this point more clearly. We also now indicate $V(t_{LZT})$ in Fig. 4a, b, which directly shows that the voltage relevant for the LZT is nearly frequency independent as argued above. In addition, in the supplementary information, we have added Fig. S8 in which we show the value of $V(t_{LZT})$ (averaged over time) for different a.c. frequencies f_{ac} to show that it is quite insensitive to the value of f_{ac} when considering the first visible Shapiro step.

It would be helpful if the authors could expand on what they precisely mean on page 5 by "intermediate transparencies...contributions to LZT experiments." Did the authors do a multi-channel modeling?

Finally, it would be nice if the authors could clarify the role of LZT. This is because highly transparent junctions can produce LZT at both zero energy as well as energy Δ (see e.g. PRB 95, 060501(R) (2017)), which then destroys the 4π periodic nature of the current.

We did not perform a fully dynamical calculation with all the modes but we studied the effect of different values of the transparency for the two-channel model that we present. We considered the situation in which one channel had a large transparency, and therefore a large P_{LZT} , and one channel had an intermediate transparency and therefore a intermediate values of P_{LZT} . For this situation we obtained a chaotic dynamic for the phase that resulted in no discernible Shapiro steps. This situation is not observed in our devices. This can be understood considering the spectrum for the type of Josephson junctions (JJs) that we study. For our JJs are the distance between the two superconducting leads L is of the order of 100 nm and the width W is $4\ \mu\text{m}$. Let n_y be the integer

labeling the transverse modes. For each transverse mode, n_y , the longitudinal Fermi velocity $v_{F,x}^{(n_y)}$ is given by:

$$v_{F,x}^{(n_y)} = \frac{2at}{\hbar} \left(\frac{\mu}{t} - \pi^2 \frac{a^2}{W^2} n_y^2 \right)^{1/2}$$

where $t = \hbar^2/(2m^*a^2)$ is the nearest neighbor hopping amplitude with m^* the effective mass, a the lattice constant, and μ the chemical potential. For the case of InAs $m^* \approx 0.03m_e$, $a = 0.6$ nm, and $\mu \approx 0.1$ eV. From $v_{F,x}^{(n_y)}$ we can then extract the effective “longitudinal” coherence length, ξ_{n_y} , of each transverse mode:

$$\xi_{n_y} = \frac{v_{F,x}^{(n_y)}}{\pi\Delta}.$$

We can then see that for $W = 4\mu\text{m}$ and $L = 100$ nm there are tens of modes for which $L > \xi_{n_y}$, i.e. modes for which the junction appears to be long. The ABSs corresponding to these modes will therefore have energy well below Δ . Figure S4 in the revised supplementary information shows the ABS spectrum for the case when $L = 100\text{nm}$ and $W = 500$ nm (it is computationally challenging to consider larger widths). We can see in Fig. S4 that that for these parameters there are at least of couple of “long-junction” modes, i.e. modes that have energy well below Δ for any value of ϕ . In our case these modes are also highly transparent and therefore have a very high probability to undergo a LZT at π . On the other hand, given their large separation from the continuum, they have completely negligible probability to undergo LZTs to the continuum. These modes are therefore responsible for the 4π signal to the current. We notice that all the “short-junction” modes have lower transparency than the “long-junction” modes and so have a lower probability to undergo a LZT at π . In addition, for $\phi = 2m\pi$ the “short-junction” modes merge into the continuum. These two facts imply that the “short-junction” modes will almost exclusively be able to contribute a 2π signal to the Josephson current. In the presence of disorder the “long-junction” modes become diffusive. In this case given the bimodal character of the distribution of transparencies in diffusive conductors, we expect that some of the “long-junction” modes will still have transparency close to 1.

This situation is analogous to the one discussed in Ref. 43 in which a resonant impurity bound state, by weakly coupling to the a high transparent mode effectively turned it into a “long-junction” mode well separated from the continuum at Δ . The presence of highly transparent, “long-junction”, modes in our devices appears to be the key for the observation of missing Shapiro steps.

In the revised manuscript we expanded the relevant discussion to better explain this point and added Fig. R1 as Figure S4 to the SI along with a discussion along the lines of the one presented above.

In summary, I think the present manuscript can be interesting as a systematic study of non-topological origin of missing Shapiro steps. At the same time, because of the muddled nature of the literature I cannot accept the claim that this is the first time that this has been seen, though it hasn't been stated before. In my view, whether this manuscript should be published in Nature Comm or a more specialized journal depends on the extent to which it can clarify this issue and characteristics of Shapiro from non-topological sources.

In the substantially revised manuscript we think we have addressed all the important issues that the reviewer had raised. We now more clearly present the results in the context of previous works and more clearly explain the significance of our findings. In addition, motivated by the reviewers'

reports we now present even more compelling experimental results that show that the observation of missing odd Shapiro steps in our devices is robust, and not due to hysteresis effects. We also present a more compelling theoretical scenario to explain the missing Shapiro steps in our junctions. We hope the reviewer will appreciate the improvements to the manuscript and find that it now more clearly presents the important result that some of the key signatures of topological superconductors have been observed in JJs unambiguously in a topologically trivial regime. We hope the reviewer will agree that this result should be of interest to the broad readership of Nature Communications, given the importance that the ability to unambiguously identify superconducting topological states has for both fundamental reasons and for the development of topological qubits.

II. REFEREE 2: REMARKS TO THE AUTHORS

We thank the referee for the thoughtful reports and for raising some important issues that we address in detail below and in the revised manuscript.

The authors report the realization of Al-InAs-Al heterostructure using MBE epitaxial deposition. They measured the d.c. and a.c. the response of the samples. A surprising suppression of the odd-number Shapiro steps was observed, which the authors attributed to the Landau-Zener transition in few Andreev bound states with high transparency. I believe that such good samples and data quality worth being published. But before I fully recommend the publication, there are few points that need to be addressed in the revised version.

1. The Landau-Zener transition probability in a highly transparent ABS is simply given by $P_{LZT} = \exp(-\pi \Delta(1 - \tau)/eV)$ (Equ. (3) in the manuscript) which suggests a higher transition probability at higher bias voltage, similarly at a higher frequency. This should suggest a stronger 4π signal (suppression of the odd steps), which is contradictory to the observed results (at higher bias voltage, the odd missing steps reappear). This was also argued in the previous works by showing a frequency dependence and the recovered odd steps at higher bias voltage. Could the authors provide an explanation within the same model?

We thank the referee for his/her comment that in conjunction with a similar comment from the first referee has prompted us to clarify the main text and the SI regarding this point.

The voltage entering the expression of the LZT probability is the instantaneous voltage in the vicinity of the anti-crossing at $\phi = \pi$, $V(t_{LZT})$. To avoid confusion it is useful to separate the dependence of $V(t_{LZT})$ on the frequency, f_{ac} , and the d.c. bias current bias I_{dc} (at a fixed value of I_{ac}).

$V(t_{LZT})$ is almost independent of f_{ac} . As we write in the revised manuscript, and also in our reply to reviewer #1, this conclusion can be recovered by considering the analogy between the dynamics of the phase, described by Eq. 1., and the damped dynamics of a massless (given that the capacitance is negligible in our devices) particle in a washboard potential. The instantaneous voltage across the junction spikes when the particle falls from one minimum of the washboard to another. This happens when the sum of I_{dc} and I_{ac} overcome the modulation of the washboard potential (I_c). The speed the particle achieves during this transition is directly related to the friction (R_n), and amplitude of the dc and ac component of the current. In the first steps, the sum of both those amplitude is typically of the order of the modulation of the washboard potential (I_c), which they need to overcome to induce a movement of the particle. It is, however, independent on the frequency since the particle is massless. As a consequence of the previous point, the P_{LZT} is independent of the frequency and the only expected frequency dependence is the fact that the suppression of the odd steps can persist only up to frequencies of the order of $f_{4\pi} \approx 2eR_n I_{4\pi}/h$ where $I_{4\pi}$ is the critical current that can be carried by the high transparency modes with large P_{LZT} as discussed in Ref. 4 and 5.

In the revised manuscript this point is explained more clearly. In addition, we now indicate $V(t_{LZT})$ in Fig. 4a, b, which directly shows that the voltage relevant for the LZT is nearly frequency independent as argued above. In addition, in the supplementary information, we have added Fig. S8 in which we show the value of $V(t_{LZT})$ (averaged over time) for different a.c. frequencies f_{ac} to show that it is quite insensitive to the value of f_{ac} .

Conversely the reviewer is correct that $V(t_{LZT})$, and therefore P_{LZT} , increases with the bias and therefore, all things being equal, more suppression of the odd Shapiro steps should be observed at larger I_{dc} . This observation is what makes necessary to enrich the model by allowing two channels: a channel with large P_{LZT} and critical current $I_{4\pi} \ll I_c$, and a channel with negligible P_{LZT} and critical current $I_{2\pi} \gg I_{4\pi}$. As the bias current is increased (resulting, in larger average voltage) more and more of the current is carried by the channel with negligible P_{LZT} that only contributes a 2π signal. When the current in the channel with negligible P_{LZT} is larger than the one in the channel with large P_{LZT} the odd Shapiro steps reappear.

This scenario is consistent with the nature of the spectrum of the JJs that we study. As shown in the new Fig. S4 the spectrum is characterized by few highly-transparent modes with energy well below Δ for all values of ϕ , and modes with lower transparency that merge to the continuum for $\phi \approx 2m\pi$. The first modes are responsible for the 4π signal whereas the latter can only contribute a 2π signal.

2. One reason that the 4π -periodic current was not observed in many highly transparent trivial systems is that the quasiparticle poisoning rate (states drop from the upper ABS branch to the lower one) is much higher in these systems. It is essential to understand why it is significantly lower in the reported system. A major cause of such relaxation is the inelastic scattering, which can be strongly suppressed in a system with strong spin-orbit coupling. In ref.[27], authors point out that two zero-energy modes can be formed due to two degenerate Dirac cones with opposite chirality, and the orthogonality between the two cones prevents the two modes from annihilating. Therefore a 4π -periodic current can be observed. A similar picture can be found in the current InAs junction, without the magnetic gap, two Fermi surfaces at different momenta should have a low inter-cone scattering rate, and give two (very near) zero-energy modes. Can the authors elaborate more on this in the discussion part?

This is also an important point. For the case of degenerate Dirac cones the degeneracy is protected by the different chirality of the cones. Such degeneracy explains the lack of hybridization between Andreev bound states (ABSs), or Majorana, but in itself does not protect against possible relaxation processes caused by quasiparticle poisoning. As the reviewer implies, and as pointed out, for instance, in Ref. 47, the relaxation between ABSs branches is strongly suppressed if the ABSs do not come into contact with the continuum as otherwise they relax immediately from the upper branch to the lower branch. In our case, the modes carrying a 4π periodic supercurrent are well separated from the continuum. As a consequence the quasiparticle poisoning of these modes is greatly suppressed. Quasiparticle dynamics have been studied in detail in metallic superconductors but have not been studied extensively in semiconductor–superconductor structures. Recent experimental work on junctions formed by Al/InAs nanowires, *Higginbotham, A. P. et al. Parity lifetime of bound states in a proximitized semiconductor nanowire. Nature Physics 11, 1017–1021 (2015)*, has extracted a quasiparticle poisoning time of the ABSs in excess of 10 ms. These considerations give us confidence that quasiparticle poisoning should not affect our measurements.

In the revised version of the manuscript we have added figure S4 in the SI to clearly show the separation of the 4π modes from the continuum. In addition we now explicitly state that such separation will protect the 4π modes from quasiparticle poisoning.

3. Please provide details of the fabrication process. E.g. the thickness of the Al layer, how many lithography steps were involved, etc.. The authors refer to the supplementary information as “Details of fabrication and measurements are described in the SI.”, but there is no description in the SI.

The Methods have been expanded to include more details regarding the fabrication process. It now reads:

The samples were grown on semi-insulating InP (100) substrates in a modified Gen II MBE system. The step graded buffer layer, $\text{In}_x\text{Al}_{1-x}\text{As}$, is grown at low temperature to minimize dislocations forming due to the lattice mismatch between the active region and the InP substrate. After the quantum well is grown, the substrate is cooled to promote the growth of epitaxial Al (111)^{29–31}. The typical thickness of the grown aluminium layer is 20 nm.

The fabrication process consists of two step of electron beam lithography using PMMA resist. After the first lithography, the deep semiconductor mesas are etched using first Transene type D to etch the Al and then a III-V wet etch solution ($\text{C}_6\text{H}_8\text{O}_7$ (1M) 18.3: H_3PO_4 (85%) 0.43: H_2O_2 (30%) 1: H_2O 73.3) to define deep semiconductor mesas.

III. REFEREE 3: REMARKS TO THE AUTHORS

We thank the referee for his/her thorough report and for pointing out some of the shortcomings of the data and analysis presented in the original version of the manuscript. Below we address in detail the issues raised by the referee and describe the changes made in the revised version of the manuscript to address these issues.

In the manuscript by Dartiailh, et al., the authors study Al/InAs/Al Josephson junctions under microwave irradiation. They map out the patterns of Shapiro steps in the current-voltage characteristics of such devices as a function of microwave power and find that the first voltage step is suppressed at low RF drive amplitude. While a similar suppression has been observed in Josephson devices with weak links made from topologically nontrivial materials, the devices under test are in the trivial regime. The authors attribute the effect to Landau-Zener transitions between Andreev bound states of channels with high transparency and go on developing a RSJ model description of the junction dynamics. They add the contribution of a single highly transparent mode to an otherwise sinusoidal current-phase relation and allow for Landau-Zener transitions between the two states of the transparent mode at odd multiples of phase π with a phase velocity dependent LZ probability. (I infer this from the reference by Dominguez, et al., since it is not clear in the text.) It is then found from the numerical solutions of the RSJ model that Shapiro steps at voltages corresponding to odd multiples of the drive frequency are suppressed at low drive amplitude (most clearly seen in Fig. S2). Based on these data the authors argue the case for 4π -periodic dynamics due to Landau-Zener transitions in their junctions. The appearance of weak half-integer Shapiro steps at higher drive frequencies is presented as additional evidence for the presence of high transmission modes in these samples.

We apologize for the lack of clarity in the introduction of the theoretical modelling. The referee properly inferred the key elements of our model and we have now clarified its presentation in the main text, which now reads:

To model the supercurrent flowing across the JJ we use two effective modes: a very low transparency mode with a purely sinusoidal CPR and completely negligible probability to undergo a LZT, and an effective mode with very high transparency τ which can undergo a LZT at $\phi = \pi$.

Shapiro step patterns with missing steps have been put forward as key evidence for (topological) midgap Andreev bound states in Josephson devices with topological insulator weak links. Critics have argued that it should be easy to replicate this effect in topologically trivial Josephson devices with (few) highly transparent modes if Landau-Zener transitions occur between the two states. I believe it to be of great value to verify such a scenario experimentally, but I am not aware of any experimental results. Dartiailh, et al., report that the first Shapiro step in the current-voltage characteristics of Al/InAs/Al Josephson junctions is missing at low microwave drive amplitude. They attribute this to a 4π -periodic Josephson effect enabled by Landau-Zener transitions. However, I find the data unconvincing. Hysteresis effects can easily be mistaken for a missing first Shapiro step. Although the authors state that this is not the case, I must disagree.

Firstly, from the tiny insert in Fig 1d, the voltage at which retrapping occurs in sample A is ca. 25 μ V. This corresponds to a frequency of 12 GHz. Below this frequency the hysteresis interferes with the visibility of the first Shapiro step. Secondly, despite the unfortunate choice of a color scheme, we can see this directly in the outlines of the dark blue zero-voltage region at low microwave power

in the colorplot, Fig. 2e/f (sample A). From the parameters in Table 1 and colorplot Fig 2g/h, I presume that the situation is similar in sample B.

In relation to this point the reviewer then concludes:

In my opinion, the experimental data in the manuscript do not support the conclusion. To make a stronger case, additional measurements are needed. The authors could, e.g., shunt the device with a parallel resistor to remove the hysteresis or repeat the measurements at higher temperatures when the critical current is smaller. I cannot recommend publication of the manuscript in its present form.

This is a central criticism that we thank the reviewer for pointing out and allowing us to address in detail. Motivated by the reviewer's report we have done a lot of additional work to check carefully if hysteresis effects were responsible for the observed missing steps. We have measured new devices, we have added a completely new figure, Fig. 3, to the manuscript, and a new figure to the supplementary material, Fig S2, to present new measurements on a new device in a configuration with no discernible hysteresis, and we have significantly expanded the discussion of hysteresis effect in the revised manuscript. Below, in the revised manuscript, and in the revised supplemental material, we summarize our new findings that allow us to much more compellingly argue that Landau-Zener transitions for highly transparent modes are responsible for the observed missing Shapiro steps.

The reviewer is correct that both device A1 (previously A) and device B present a marked hysteresis with a retrapping voltage higher than the first step. Motivated by the reviewer's comments we carefully reviewed and now clearly report in the revised manuscript the retrapping voltage for device A1. We have that in device A1 the retrapping voltage is $36 \mu\text{eV}$. This value is larger than the potential corresponding to the 4th Shapiro step at 4 GHz. If hysteresis effects were to wash away the Shapiro steps one would expect that for such value of the retrapping voltage not only the 1st Shapiro step would be missing, but also the 2nd, and possibly the 3rd and 4th, would be suppressed as was for example recently reported in Ref. 40. However, this is not what we observe. This observation gives us some confidence that hysteresis effects alone might not be the main reason for the missing 1st Shapiro steps but this is only weak evidence of the limited impact of hysteresis on our measurement, and so we took the time to take new measurements for devices and conditions in which there is no discernible hysteresis as the reviewer also suggested in the concluding remark of his/her report. We hence considered a new device, device A2 in the revised manuscript, that was fabricated on the same piece of wafer and at the same time as device A1 and measured it at high temperature to minimize its hysteresis. The new figure 3a shows that for device A2, for temperatures larger than 600 mK, the hysteresis becomes negligible. The new figure 3b shows the VI for device A2 taken at 650 mK: we see that at this temperature device A2 shows no discernible hysteresis. We then measured the IV curves of device A2 at 650 mK in the presence of microwave radiation. The results are presented in the new Figures 3c-g and Fig. S2a-f and discussed in the new section, "Experiments at higher temperature", of the revised manuscript. These results show that even when no hysteresis is present, the first Shapiro step is missing and allow one to rule out hysteresis effects as the dominant source in our devices for the observed missing Shapiro step.

We hope the reviewer will appreciate the work we have done to address as thoroughly as possible the important point that he/she raised in his/her report and that he/she, and the readers, will find the additional measurements, new figures, and discussion included in the revised manuscript sufficiently compelling to conclude that Landau Zener transitions of highly transparent modes are the most likely source of the observed missing Shapiro step in our devices.

Additionally, the authors point out a weak suppression of the third step in Fig 2g/k. Here I urge caution because resonances in the device or measuring circuit wiring may wash out one or more of the steps. The Shapiro step patterns should be studied carefully for several (nearby) frequencies and in a larger range of microwave powers to rule out such spurious effects.

Prompted by the referee comment, we re-examined our data. We noticed that the data presented in the Supplementary Information Fig. S1 at 4 and 5 GHz present the same trend. To better illustrate this point, Fig. S1 now includes an histogram of the voltage for one microwave power. With this, we show that the suppression of the third step is visible over at least 2 GHz. We lack data for several nearby frequencies and device B was not available for additional measurement, however observing the same signature for these three frequencies, which were not hand picked, allow us to reasonably rule out spurious effect related to the setup. We now highlight this fact in the main text.

In addition, in the case of sample B, a weak suppression of the third Shapiro step is visible in Fig. 2g and k. This signature is also present in the data presented in SI Fig. S1 at 4 and 5 GHz on the same sample allowing us to rule out spurious experimental setup effects.

IV. SUMMARY OF CHANGES

- Prompted by the referee #1 comment we have largely rewritten the introduction to better underline the specificity of our work in the context of the existing literature on missing Shapiro steps in topological Josephson junctions. As part of this re-writing we have addressed the concerns of the referee #1 regarding our references to the existing literature.
- To answer the concern of referee #3 we have performed additional measurements on a third device in a configuration with no discernible hysteresis. Our results are presented in a completely new figure, Fig. 3, in the manuscript, and a new figure to the supplementary material, Fig S2 which confirm and reinforce our previous findings. We have also significantly expanded the discussion of hysteresis effect in the revised manuscript.
- To answer the concern of referee #3 regarding the suppression of the third Shapiro step in sample B, we have added histogram to Fig. S1 to better illustrate the suppression of the third step also at 4 and 5 GHz.
- To address the comments of both referee #1 and #2 we have substantially reworked the theory discussion both in the main text and in the SI.
 - First, we clarified the specificity of our system, namely the presence of high transparency "long-junction" modes that are well separated from the continuum. This separation from the continuum can in particular explain why the system does not relax at 2π . The discussion of the main is supported in the SI by a tight-binding simulation presented in Fig. S4.
 - The above discussion allowed us to better justify our choice of a model comprising few highly transparent modes that can undergo a LZT at $\phi = \pi$ and a large fraction of purely 2π periodic modes.
 - Finally we clarified the importance of considering the instantaneous voltage at $\phi = \pi$, $V(t_{\text{LZT}})$ when computing the LZT probability. We now provide a simple argument in the main text, based on the dynamic of a particle in a tilted washboard potential, to explain why $V(t_{\text{LZT}})$ is independent of the frequency in the condition of our experiments. We also updated the theoretical figure of the main text to clearly indicate the value of $V(t_{\text{LZT}})$, making it easier to the reader to notice the frequency independence. Those results are also supported in the SI by Fig. S6 and the new S7.
 - To avoid making the SI too long, we merged the two figures presenting the numerical results for the samples parameters in one presenting only the histograms since the line-cuts are shown in the main text.
- As a consequence of the addition of the new figure 3, we have moved what was figure 4 and the related discussion to the SI.

Reviewer #1 (Remarks to the Author):

Dear Editor,

The authors of the manuscript titled "Missing Shapiro steps in topologically trivial Josephson Junction on InAs quantum well" have responded to comments by the three referees. Specifically, in response to my central criticism i.e. the novelty compared to incidental evidence in earlier papers, the authors made the valid point that since the experiments were designed to establish topological states - they could not definitely establish that the system was non-topological. They solidified their point by referring to a recent paper Phys. Rev. Lett. 124, 076802 (2020)). The authors also responded to a theoretical issue about the voltage dependence by referee 2 and most importantly they provided data for devices with minimal hysteresis in response to referee 3. The point by referee 3 about hysteresis and resonances is a key challenge in the interpretation of Shapiro steps. It looks to me that Referee 3's comments have been responded to - but I would defer to Referee 3 to decide this.

Assuming that referee 3 believes that the issues of hysteresis and resonances have been established, I am happy to recommend this manuscript for publication in Nature Communications.

Reviewer #2 (Remarks to the Author):

After reading the reports, I believe that the authors replied properly to the questions and comments. I believe that this paper is suitable as it is for publication in Nature Communications.

Reviewer #3 (Remarks to the Author):

In the revised manuscript, the authors present additional measurements at a higher temperature. Under these conditions, it is unlikely that the absence of a first Shapiro step is an artifact due to hysteretic switching. The data in Fig. 3 strengthens the conclusion that a 4π -periodic Josephson current component is present in the device. Conversely, it was suggested by Referee 1 that the present work is not novel, pointing to Pribyl et al, Nat. Nano. 10, p. 593-597 (2015). In that work, however, the first Shapiro step is only missing for positive source-drain current which is a clear case of hysteretic switching interfering with the effect. When the device is in the voltage state (on the negative branch of the IV characteristic), the first Shapiro step appears to be fully formed. Thus, it remains unclear to me whether Pribyl et al. have observed the effect.

There are some unlinked references in the Supplementary Information.

All my concerns have been addressed by the authors. I recommend publication.